Insight into the kinematics of blue whale surface foraging through drone observations and prey data

Torres Leigh G. leigh.torres@oregonstate.edu 1
Barlow Dawn R. 1
Chandler Todd E. 2
Burnett Jonathan D. 3
1 Geospatial Ecology of Marine Megafauna Lab, Marine Mammal Institute, Department of Fisheries and Wildlife, Oregon State University , Newport , OR , United States of America
2 Geospatial Ecology of Marine Megafauna Lab, Marine Mammal Institute, Oregon State University , Newport , OR , United States of America
3 Aerial Information Systems Laboratory, Forest Engineering, Resources and Management, Oregon State University , Corvallis , OR , United States of America
Boyer Alison
Electronic publication date: 2020 Apr 22
Publication date: 2020
Volume: 8
Electronic Location ID: e8906
Received 2019 Oct 18; Accepted 2020 Mar 12
Copyright: ©2020 Torres et al.
Copyright year: 2020
Copyright holder: Torres et al.
License: This is an open access article distributed under the terms of the Creative Commons Attribution License, which permits unrestricted use, distribution, reproduction and adaptation in any medium and for any purpose provided that it is properly attributed. For attribution, the original author(s), title, publication source (PeerJ) and either DOI or URL of the article must be cited.
License URL: https://creativecommons.org/licenses/by/4.0/

Keywords: Blue whale, Optimal foraging theory, Krill, Unmanned aerial systems, Foraging ecology, Predator-prey interactions, Energetics, Surface lunge feeding, New Zealand, Prey response

Funding: The Aotearoa Foundation The New Zealand Department of Conservation The Marine Mammal Institute at Oregon State University Greenpeace New Zealand, OceanCare, Kiwis Against Seabed Mining The International Fund for Animal Welfare The Thorpe Foundation Funding for this research was provided by The Aotearoa Foundation, The New Zealand Department of Conservation, The Marine Mammal Institute at Oregon State University, Greenpeace New Zealand, OceanCare, Kiwis Against Seabed Mining, The International Fund for Animal Welfare, and The Thorpe Foundation. The funders had no role in study design, data collection and analysis, decision to publish, or preparation of the manuscript.

==============================
To understand how predators optimize foraging strategies, extensive knowledge of predator behavior and prey distribution is needed. Blue whales employ an energetically demanding lunge feeding method that requires the whales to selectively feed where energetic gain exceeds energetic loss, while also balancing oxygen consumption, breath holding capacity, and surface recuperation time. Hence, blue whale foraging behavior is primarily driven by krill patch density and depth, but many studies have not fully considered surface feeding as a significant foraging strategy in energetic models. We collected predator and prey data on a blue whale (Balaenoptera musculus brevicauda) foraging ground in New Zealand in February 2017 to assess the distributional and behavioral response of blue whales to the distribution and density of krill prey aggregations. Krill density across the study region was greater toward the surface (upper 20 m), and blue whales were encountered where prey was relatively shallow and more dense. This relationship was particularly evident where foraging and surface lunge feeding were observed. Furthermore, New Zealand blue whales also had relatively short dive times (2.83 ± 0.27 SE min) as compared to other blue whale populations, which became even shorter at foraging sightings and where surface lunge feeding was observed. Using an unmanned aerial system (UAS; drone) we also captured unique video of a New Zealand blue whale’s surface feeding behavior on well-illuminated krill patches. Video analysis illustrates the whale’s potential use of vision to target prey, make foraging decisions, and orient body mechanics relative to prey patch characteristics. Kinematic analysis of a surface lunge feeding event revealed biomechanical coordination through speed, acceleration, head inclination, roll, and distance from krill patch to maximize prey engulfment. We compared these lunge kinematics to data previously reported from tagged blue whale lunges at depth to demonstrate strong similarities, and provide rare measurements of gape size, and krill response distance and time. These findings elucidate the predator-prey relationship between blue whales and krill, and provide support for the hypothesis that surface feeding by New Zealand blue whales is an important component to their foraging ecology used to optimize their energetic efficiency. Understanding how blue whales make foraging decisions presents logistical challenges, which may cause incomplete sampling and biased ecological knowledge if portions of their foraging behavior are undocumented. We conclude that surface foraging could be an important strategy for blue whales, and integration of UAS with tag-based studies may expand our understanding of their foraging ecology by examining surface feeding events in conjunction with behaviors at depth.

Introduction

Optimal foraging theory predicts that predators maximize energetic gain by choosing to exploit prey so that caloric intake outweighs energetic costs of search, capture and digestion (Charnov, 1976). Foraging marine mammals must also include oxygen consumption, breath holding capacity, and surface recuperation time in this energetic balance. Yet, application of optimal foraging theory in marine systems can be challenging due to often obscured knowledge of prey availability and predator behavior. As the largest predator on earth, the blue whale (Balaenoptera musculus) has massive prey requirements to meet energy demands (Williams et al., 2001). Hence, their foraging ecology must be efficient and optimized through adaptable feeding strategies primarily relative to prey patch depth and density (Croll et al., 2001; Doniol-Valcroze et al., 2011; Goldbogen et al., 2011; Goldbogen et al., 2013a; Goldbogen et al., 2015; Hazen, Friedlaender & Goldbogen, 2015; Friedlaender et al., 2016), while also maintaining dependence on the surface for oxygen supply. This tie to the surface has led to the successful application of central place foraging theory (Orians, 1979) to blue whale foraging behavior, with both surface recuperation and foraging times increasing with target prey patch depth (Doniol-Valcroze et al., 2011).

Blue whales are rorqual whales that employ a lunge feeding method where the whale accelerates forward rapidly and opens it jaws to engulf prey-laden water into an extensible buccal cavity that is then filtered through baleen plates (Goldbogen et al., 2017). Lunge feeding is energetically demanding due to body acceleration, engulfment and acceleration of water mass, and drag forces, all of which deplete oxygen stores and limit dive duration (Acevedo-Gutierrez, Croll & Tershy, 2002; Potvin, Goldbogen & Shadwick, 2010; Goldbogen et al., 2012). Hence, for lunge feeding to be energetically efficient whales must locate and engulf high-density prey patches to maximize energy gain. Blue whales lunge feed at depths ranging from the surface to over 300 m (Doniol-Valcroze et al., 2011; Hazen, Friedlaender & Goldbogen, 2015) and are specialized predators of krill, which are ephemeral and patchy (Schoenherr, 1991; Croll et al., 2005; Goldbogen et al., 2011; Hazen, Friedlaender & Goldbogen, 2015; Nickels, Sala & Ohman, 2019). Therefore, blue whales must forage selectively by targeting dense prey aggregations that compensate for the energetic cost of lunging and diving (Doniol-Valcroze et al., 2011; Hazen, Friedlaender & Goldbogen, 2015).

The ecology and energetics of blue whale foraging has been illuminated through multiple studies that analyzed data acquired from animal-borne tags applied to whales in southern California, USA (Goldbogen et al., 2011; Goldbogen et al., 2012; Goldbogen et al., 2015; Hazen, Friedlaender & Goldbogen, 2015; Friedlaender et al., 2016). These studies demonstrate that blue whales modulate lunge feeding rates as a function of prey patch density and depth to optimize energetic efficiency (Goldbogen et al., 2011; Hazen, Friedlaender & Goldbogen, 2015). A detailed kinematic analysis of blue whale lunge feeding estimated the energetic costs, gains, and efficiency of a single lunge at the surface and multiple lunges at depth (Goldbogen et al., 2011). In both scenarios energy gained from engulfed krill exceeded energetic costs, yet results indicate that a single surface lunge is 2.5 times more energetically efficient than multiple (3.5) lunges conducted during a dive to depth (200 m). Hence, based on optimal foraging theory, a blue whale should only feed at depth when relative krill density at depth exceeds surface density so that the energetic costs of diving are offset. Additionally, models of blue whale foraging efficiency based on tag data demonstrated that foraging at depth only exceeds the net energy gain of surface feeding when krill density at depth is >3 times surface densities (Goldbogen et al., 2011). This scenario may be typical of target prey in the southern California ecosystem (Euphausia pacifica; Thysanoessa spinifera) that demonstrate strong diel cycles (Croll et al., 2005) and association with bathymetric features (Schoenherr, 1991; Nickels, Sala & Ohman, 2019), but possibly not of other krill species targeted by blue whales in other ecosystems.

Blue whale surface lunge feeding (SLF) is commonly observed in many ecosystems (Schoenherr, 1991; Gill, 2002; Doniol-Valcroze et al., 2011; Kot et al., 2014; Buchan & Quiñones, 2016), and may be an important foraging strategy for blue whales as an alternative to energetically demanding deep foraging, particularly in ecosystems where krill are distributed more evenly vertically through the water column or biased toward the surface. Indeed, when lunges at depth and the surface were both assessed in tag data and incorporated into a central place foraging model applied to blue whales feeding in the St. Lawrence River, Canada, predicted dive efficiency was highest at the surface and declined steadily with depth (Doniol-Valcroze et al., 2011). High rates of surface foraging by humpback whales (Megaptera novaeangliae) have also been documented (Friedlaender et al., 2009; Ware, Friedlaender & Nowacek, 2011; Owen et al., 2017), even when prey density was greater at depth (Goldbogen et al., 2008). While rorqual surface feeding has been observed and reported, quantitative description and comparison of whale surface kinematics are often excluded from accelerometer tag data analysis due to biomechanical processes and forces that restrict accurate algorithmic detection of lunges at the surface (Allen et al., 2016; Owen et al., 2016). Hence, biomechanic description of lunge feeding has focused on deep foraging dives and surface lunging is relatively poorly understood (Allen et al., 2016).

In New Zealand and southern Australia, pygmy blue whales (B. m. brevicauda) feed on a coastal krill species (Nyctiphanes australis) known for its surface swarming behavior and lack of diel or depth patterns relative to density and biomass (O’Brien, 1988; Young et al., 1993). Given the energetic costs of diving, and shallow or homogenously distributed prey, blue whales in these ecosystems should preferentially feed near the surface. Some evidence for this expectation is available from southern Australia where krill surface swarms were associated with 48% of blue whale sightings, with frequent observation of surface and sub-surface feeding from aerial surveys (Gill, 2002). Thus, disregard of blue whale surface feeding may bias our understanding of their foraging ecology and hinder conservation efforts in some ecosystems. Due to the context dependence of blue whale response to disturbance events based on behavioral state (Goldbogen et al., 2013b) and prey availability (Friedlaender et al., 2016), efforts to de-couple the 3-D overlap of whales and anthropogenic impacts should consider the functional habitat use of blue whales through the whole water column. For instance, blue whales demonstrate an increased response to shipping traffic when at the surface (McKenna et al., 2015).

Blue whale SLF events are a dramatic predation spectacle leading to multiple descriptions (Fiedler et al., 1998; Corkeron, Ensor & Matsuoka, 1999; Gill, 2002; Kot et al., 2014; Buchan & Quiñones, 2016), but surface feeding behavior has rarely been incorporated into our understanding of their optimal foraging strategies, with the exception of Doniol-Valcroze et al. (2011) and Goldbogen et al. (2011), both of which indicate the high net energetic gain of surface feeding. Therefore, incomplete quantification of surface feeding behavior, either deliberate or due to data limitations, can limit or bias understanding of how blue whales make optimal foraging trade-offs between prey depth and density, which are likely a function of ecological context and allometric scale. In this study, we examine concurrent data on krill availability and blue whale distribution within a newly described foraging area in New Zealand (Torres, 2013; Barlow et al., 2018) to examine the relationships between prey depth and density and whale dive times and behavior. We consider these data within previously described blue whale foraging efficiency (Goldbogen et al., 2011) and optimality (Doniol-Valcroze et al., 2011) models, and posit that surface feeding among these relatively diminutive blue whales in this ecosystem is a significant foraging strategy that maximizes their energetic gain. Additionally, we analyze four blue whale surface foraging events captured via Unmanned Aerial System (UAS; drone) to demonstrate the utility of drones to provide novel and detailed data on whale kinematics, sensory use, and decision making. We provide a quantitative kinematic description of a blue whale ‘lateral lunge feeding’ surface event (Goldbogen et al., 2006; Kot et al., 2014), where the whale rotates >90 ∘ along its longitudinal axis to engulf a krill patch, and compare this description to reports of blue whale lunge feeding events at depth recorded via tag data (Goldbogen et al., 2011; Goldbogen et al., 2013a; Goldbogen et al., 2015; Cade et al., 2016; Friedlaender et al., 2017). Additionally, this footage is unique due to the well-illuminated krill patch, allowing assessment of predator and prey simultaneously. We suggest that UAS can complement tagging efforts by providing quantitative data on surface foraging behavior from a new perspective (Torres et al., 2018), allowing a more complete description of blue whale foraging strategies.

Materials & Methods

Data collection

Fieldwork was conducted in February 2017 on a blue whale foraging ground in the South Taranaki Bight (STB) region of New Zealand (Fig. 1; Barlow et al., 2018) where whales target aggregations of N. australis. Blue whale occurrence data was collected during standardized survey effort (Barlow et al., 2018) aboard a 19.2 m vessel at speeds of 8 to 12 knots in suitable weather conditions (Beaufort Sea State < 5). Survey observation data was collected at 4 m above the waterline by two observers scanning either the port or starboard sides of the trackline independently. Survey tracklines did not follow a standardized layout (i.e., sawtooth or grid pattern lines), but rather aimed to maximize encounter rates with blue whales and therefore had an irregular trackline pattern directed toward productive (based on remotely sensed images of chlorophyll-a and sea surface temperature) or previously unsurveyed areas. Survey effort stopped at all sightings of blue whales and the whale(s) was slowly approached for behavioral observation. Whale behavior states were classified as travel, forage, social, rest, or unknown based on observations, with further details, such as observations of SLF events, noted. Travel was defined as directional movement and regular surfacing. Indications of foraging included surface lunges and staying in one area for a prolonged period with irregular surfacings or fluke-out dives. Social behaviors included mother-calf nursing, prolonged coordinated surfacing such as racing, and tactile contact between individuals. Resting behavior consisted of logging near the surface with minimal forward movement. All behaviors that did not fit within these classifications were considered unknown. Dive times of individual whales at all sightings were recorded as the interval of time between the whale’s surfacings. The mean dive time was attributed to the sighting. If multiple individuals were present, dive times were recorded for each individual whale, with an average dive time calculated per whale, and then a group mean was determined for that sighting. During survey effort and at many whale sightings, krill distribution data was collected by a Simrad EK60 echosounder. Field research on blue whales was conducted under research permit approval from the New Zealand Department of Conservation (45780-MAR) and Institutional Animal Care and Use Committee exemption from Oregon State University (16-1083).

Figure 1 Blue whale survey tracklines and sighting locations.

Survey tracklines in 2017 in the South Taranaki Bight (STB) with locations of blue whale sightings, and where surface lunge feeding was observed, denoted. Inset map shows location of the STB within New Zealand.

A DJI Phantom 4 Advanced (non-Pro) UAS was flown over blue whales using manual flight control and real time camera output for the primary purpose of body morphometric assessment through photogrammetry (Burnett et al., 2019). The UAS camera had a 3.61 mm focal length and a 0.0015 mm pixel size. On 20 February 2017 at 19:21 local time the UAS was hand-launched off the research vessel to fly over a single blue whale. Conditions were exceptional, with no wind or swell, and bright off-angle lighting, which enhanced water clarity and contrast so that multiple surface patches of krill were identifiable. The UAS was navigated toward the whale’s location after collecting calibration data over a board of known length at multiple altitudes for the purpose of later correcting the barometric altimeter (Burnett et al., 2019). Just after the whale was localized in the UAS video with the camera pointing nadir, the whale began a SLF event (Table 1, Event 1 at 19:23:38). The UAS operator navigated the drone to film the event as best as possible given the rapid sequence of events, yet unfortunately the expanded buccal cavity is out of frame for ∼1.5 s during engulfment of the krill. Although the whale dove out of sight after this feeding event, we later filmed its surface activity for another 5:47 min using the UAS. The UAS maintained an altitude between 29 and 38 m above the whale during these observations while we recorded three other instances of this same blue whale recognizing surface krill patches but not feeding (Table 1). Low UAS battery power and the setting sun forced retrieval of the UAS. A small tissue biopsy sample was then collected from the whale using a lightweight biopsy dart (Krützen et al., 2002) for genetic analysis to determine sex (Barlow et al., 2018).

Krill data analysis

Hydroacoustic data were collected using a Simrad ES120 splitbeam transducer with a 120 kHz receiver, 250 W, 1.024 ms pulse length, and 0.5 s ping rate. The transducer was mounted on a pole and deployed 1.45 m below the surface. Raw data were processed and extracted using the MATLAB-based program ESP3 (Ladroit, 2017), developed for fisheries hydroacoustic analysis. Volume backscattering (Sv) measurements were binned vertically into 1 m depth bins and horizontally into 5-ping bins. The upper 2 m of the water column were excluded due to noise from disturbances at the surface such as wind and swell, and sections with recognizable interference from CTD casts or missed pings were removed. Following complete visual inspection of the echograms for quality control, measurements made at 2 m or deeper were retained for analysis. The echosounder was not calibrated, and therefore all values for backscatter strength represent a relative characterization of prey availability within this ecosystem.

Table 1 Description the four blue whale (B. m. brevicauda) surface events filmed via Unmanned Aerial Systems (UAS) on 20-Feb-2017 in the South Taranaki Bight, New Zealand, including comparative metrics of speed, and krill patch size and response to whale.

Event	Local start time	Whale behavior	Krill patch size (m2)	Distance (m) at which whale responds to krill patch (time from intersection/strike)	Maximum speed(m s−1)	Distance (m) between krill patch and whale when krill first show flee response	Speed (m s−1) of whale when krill respond	Distance from eye to krill patch (m) when whale visually inspects patch (time from intersection/strike)	
1	19:23:38	Whale engages in surface lunge feeding event	45	20.74 (9.0)	3.33	2.00	3.00	18.46 (6.5)	
2	19:30:19	Whale flares pectoral fins to alter course slightly as she investigates a small krill patch and surfaces - no feeding	1	17.87 (6.45)	2.75	NA	NA	11.35 (2.44)	
3	19:36:36	Whale surfaces and makes a slight course change near small krill patches - no feeding	2.47	5.02 (NA)	NA	4.93	NA	8.56 (NA)	
4	19:38:05	Whale flares pectoral fins, changes speed, and rotates her body to examine a medium size krill patch - no feeding	4.72	12.29 (4.83)	2.90	3.77	2.69	15.24 (4.5)	

Zooplankton-like schools were identified by excluding ping data with volume backscattering strength below the –90 dB threshold at 120 kHz. We are confident that this threshold reliably identified and captured aggregations visible on the echosounder, thereby effectively characterizing schools within this ecosystem. The threshold value applied is more conservative than those used in a similar analysis for larger krill species with established cutoff frequencies (e.g., Euphausia superba, Bernard & Steinberg, 2013). Recognizing that we were not able to definitively exclude other ensonified species (i.e., fish) and in the absence of a known target strength for the krill species of interest (N. australis), we assume that the aggregations identified acoustically were predominately comprised of krill. Krill aggregations were identified following the methods described by Bernard et al. (2017). Each element in the acoustic matrix that qualified as krill was considered to be part of an “aggregation” if one of its eight neighboring elements also qualified as krill (Lawson et al., 2008; Bernard & Steinberg, 2013).

The mean Sv, depth, and vertical thickness were calculated for each krill aggregation (Supplementary Information, Table S1, Fig. S1). Mean depth was defined as the depth at the midpoint of each aggregation. Mean Sv of each aggregation was used to represent patch density. Aggregations were defined as being at a blue whale sighting if the center of the aggregation was within a 2 km radius of the blue whale sighting, and all other krill aggregations were considered not to be at a blue whale sighting. A threshold of 2 km was applied because blue whale sightings ranged up to 2 km from the initial sighting location; hence, this distance was considered to best characterize the availability of krill aggregations to the whale(s) at a sighting.

Krill patch density observed during the four UAS filmed events could not be quantitatively determined because the underwater portion of the patches are indiscernible. Yet, density of surface swarms of N. australis can range between 3,000 and >450,000 ind m−3, with individual krill lengths 15–18 mm and the wet weight biomass ranging between 40 g m−3 and 7 kg m−3 (O’Brien, 1988). Furthermore, O’Brien (1988) estimated the wet weight of a 15 m cigar-shaped patch of N. australis to be 100 kg. The length of the krill patch consumed during the filmed SLF event was 16.2 m (excluding the long tail on the bottom-right: Fig. 2H). Therefore, although we cannot quantify the density or biomass of the patch targeted during the SLF event, both metrics may be relatively high.

Figure 2 Body kinematics, and corresponding still images, during blue whale surface lunge feeding event derived from Unmanned Aerial Systems (UAS) image analysis.

(A) Mean head inclination and roll (with CV in shaded areas), (B) relative speed and acceleration, and (C) distance from the tip of the whale’s rostrum to the nearest edge of krill patch. Blue line on plots indicate when krill first respond to the predation event, and the purple dashed lines indicate strike at time = 0. The orange lines indicate the time at which the whale’s gape is widest, head inclination is maximum, and deceleration is greatest. (D) Image of whale eyeing krill patch with measured distance between eye and patch. (E) Image taken when krill begin to respond to predation event. (F) Image illustrating krill flee response. (G) Image of widest gape and angle measurement. Comparative images of krill patch size and density (H) pre- and (I) post-strike. Comparative images of head inclination (J) 2 s pre-strike, (K) at strike, and (L) post-strike of surface lunge feeding event (Example roll images in Fig. S2). Images in D–L are cropped to enhance illustrations (raw video available in the Supplemental Information). All images captured at 29.5 m altitude.

UAS video of analysis of blue whale

Despite not having a laser altimeter on-board our UAS, accurate scaling between pixels and metric units of images captured in nadir was achieved using our calibration board (Dawson et al., 2017; Burnett et al., 2019). Barometric altitude was corrected for bias using pixel length (from video) and metric length measurements of the board at a range of altitudes to create a correction model (Burnett et al., 2019). Measurements were scaled from pixels to metric units by multiplying pixel measurements by the ground sampling distance (GSD) where GSD was calculated by multiplying model corrected barometric altitude by camera sensor pixel size in mm and dividing by camera focal length in mm (Comer et al., 1998).

In all four events, estimates of the whale’s position relative to the krill patch were reconstructed from the UAS video sequence using Adobe Photoshop CC (V 2015.5; hereafter Photoshop). At 0.5 s intervals, the following metrics were estimated: horizontal distance between the whale’s rostrum tip and the krill patch, instantaneous speed of the whale, instantaneous acceleration of the whale, and the whale’s body roll and head inclination angle. Also estimated were the distance at which the krill responded to the whale, krill patch area pre-strike, and krill patch area post-strike for the SLF event. The lack of reference points in the relatively homogenous ocean scene was a challenge for data analysis, however we attempted to mitigate this issue by using a consistent point on the leading edge of the krill patch as a reference marker (anchor point) and limiting data analysis to when the whale and krill were simultaneously visible. We therefore assume that the krill patch did not move significantly relative to the whale’s larger magnitude movements until its flee response from the whale at close range, which was also measured using the same anchor point to ensure consistency. We also assume no impact of water current on data analysis, which is considered a valid assumption as minimal lateral movement of the krill was observed during periods when the aircraft was stationary. Our estimated values of relative acceleration and speed should not be considered absolutes due to the many sources of minor uncertainty in UAS imagery that are difficult to quantify given the lack of reference points, in situ water current measurements, and ability to estimate error. However, our methods provide reasonable estimates of whale kinematics as confirmed by similarities to tag-derived metrics of blue whale lunge biomechanics, provide rare estimates of krill prey flee response, and are consistent across the four UAS filmed foraging events thus providing reliable relative comparisons between events. For future applications of UAS to film whale surface behavior we suggest the use of a laser altimeter to improve precision and accuracy of altitude estimates, and thus speed and acceleration measurements.

During the SLF event, time-stamps of fluke beats, pectoral fin movements, and mouth opening were also noted, and maximum gape was calculated using the online tool at https://www.ginifab.com/feeds/angle_measurement/. The times determined as when the whale first responds to the krill patch, krill begin flee response from the whale, and widest gape during the SLF event are recognized as subjective choices, yet all authors estimated these values independently and were in agreement.

The interval of interest in Event 1 when the SLF event was filmed was isolated using the video timeline functionality in Photoshop to the time frame beginning immediately before the whale changes directions in response to the krill patch and ending at the completion of the dive following the SLF event. This interval limited the analysis to frames between 135.37 s to 166.87 s and corresponds to 12 seconds pre-strike through 19.5 s post-strike. Video of the lunge feeding event was segmented into 3,840 ×2,160 pixel images at 0.5 s intervals using the video rendering function in Photoshop. Images were distortion corrected in MATLAB™ (Burnett et al., 2019) and mosaicked in Photoshop. To preserve image scale of the mosaic, a new 15,000 by 15,000 pixel blank image was generated with the same dots per inch (DPI) as the distortion-corrected images. Estimation of the whale’s relative speed and acceleration during the SLF event was limited to the time interval where the krill patch and whale were in the frame simultaneously from 9 s pre-strike to 19.5 s post-strike. These 0.5 s interval images were imported and spatially arranged using the krill strike point as the anchor position. Whale position with respect to time was estimated by recording the x, y coordinates of the tip of the whale’s rostrum position in each of the constituent images within the mosaic.

Distance at 0.5 second intervals was estimated using the x, y position at the beginning and end of each interval as inputs to the Euclidian distance equation. Distances were scaled to units of meters using the GSD equation (Burnett et al., 2019). The 3.61 mm focal length camera on a Phantom 4 Advanced at 29.5 m mean corrected altitude above-sea level resulted in a 0.0119 m GSD per pixel scaling constant. A spline function of distance with respect to time was developed by accumulating distance over time using the ‘Spline’ function in R (V 3.4.3) (R Development Core Team, 2018). The spline function allowed for the estimation of instantaneous speed and instantaneous acceleration. A spline was selected instead of using raw numbers to estimate average speed and acceleration over small time intervals in an effort to reduce the inherent noisiness of the data due to small errors associated with scaling and imprecision associated with visual interpretation. Speed was estimated instead of velocity because we were interested in accumulated distance instead of vector-dependent total displacement. Distance between the whale and krill patch was estimated by calculating the scaled pixel distance between the tip of the whale’s rostrum position and initial krill strike point at each time interval.

Following the mosaic, images associated with the undisturbed pre-strike krill patch were merged to facilitate a contiguous area estimate (Fig. 2H). Post-strike residual krill patch area was estimated using the frame at 151.87 s that displays the entire residual krill patch (Fig. 2I). Scaling of pixel areas to square meters occurs using the 0.0119 m scalar.

Individual images at 0.5 s intervals during Event 1 were visually assessed by all four co-authors independently to subjectively assess roll and head inclination throughout the SLF event. Body roll analysis began at 12 seconds pre-strike and was evaluated using the right-hand rotation convention where rotation to the left is negative and rotation to the right is positive (Fig. S2). 0° body roll occurs when the whale is swimming straight with the body level in the water. The position of the whale’s spine, pectoral fins, jaw line, and body flanks relative to the overhead perspective of the UAS camera in nadir were used to estimate the whale’s body roll. Head inclination is the measure of angular head position deviation from neutral where neutral (0°) occurs when the body is fully elongated with a neutral spine and an imaginary straight line extends from the tip of the rostrum to the tail. Head inclination ranges from 90° to −90° where positive is a position where the head is above the imaginary line connecting the rostrum and tail, and negative is the position below the imaginary line connecting the rostrum and tail. Change in head inclination is best observed at roll angles with absolute values between 45° and 110°, so estimations were limited to between 4 s pre- to 7.5 post-strike (unless unobservable between 2.5 and 3.5 s post-strike when the whale was off-screen; Figs. 2A, 2J–2L).

Subjective visual estimation of head inclination and roll angles by each co-author is imprecise but indicative of relative body attitude changes with respect to time. Interpretation of head inclination and roll was limited to 5° and 10° increments, respectively, to reflect the imprecision of the estimates (Fig. S2). Moreover, we calculated the Interrater Correlation Coefficient (ICC; Koo & Li, 2016) between the four independent evaluations of head inclination and roll by co-authors to evaluate consistency and absolute-agreement as metrics of confidence in estimations. ICC estimates and their 95% confident intervals were calculated using the ‘PSYCH’ statistical package in R based on a mean-rating (k = 4 raters), estimating consistency and absolute-agreement, using a 2-way mixed-effects model. The mean and CV values of the four head inclination and roll estimates for each 0.5 s interval image were used in Fig. 2A to illustrate the blue whale’s body positioning throughout the SLF event, and any variation in estimations.

Results

Distribution of blue whales and krill

During 887 km of survey effort in the South Taranaki Bight (STB) of New Zealand, 32 blue whale sightings were recorded of 68 individuals (Fig. 1). Foraging behavior was apparent at 15 sightings, with SLF observed at six of those sightings. Echosounder data were recorded for 90.6 h during the field season and 2,911 krill aggregations were identified from analysis of these data. The highest krill density occurred in the upper 20 m of the water column, and blue whale sightings with perceived foraging behavior occurred in habitats with greater krill density relative to background availability (Fig. 3). Furthermore, the mean depth of krill aggregations was shallower at all blue whale sightings as compared to background availability, and depth became progressively shallower when compared to sightings where foraging behavior was detected or SLF was observed (Table 2, Fig. 3). The dive times of New Zealand blue whales followed this same trend, with relatively short dive times recorded for all sightings (2.83 ± 0.27 SE min), which became even shorter at apparent foraging sightings and where SLF was observed (Table 2). Krill aggregation depth and density during the sighting of the UAS filmed whale are representative of the general pattern of krill availability across the study region (Fig. 3), thus supporting the interpretation of these filmed surface behaviors relative to the study area and whale population.

Figure 3 Density contours comparing the depth and density (Sv) of krill aggregations at blue whale foraging sightings (red shading) and in absence of blue whales (grey shading).

Density contours: 25% = darkest shade, 75% = medium shade, 95% = light shade. Blue circles indicate krill aggregations detected within 2 km of the sighting of the UAS filmed surface foraging whale analyzed in this study.

Table 2 Comparison of krill (Nyctiphanes australis) aggregation depth in the South Taranaki Bight, New Zealand, not within 2 km radius of blue whale (B. m. brevicauda) sightings and those within 2 km of sightings.

The mean dive time of whales at each sighting by group (all, foraging observed, surface lunge feeding observed) are also given.

	Sample size (blue whale sightings/ krill aggregations)	Mean krill depth (m); (SE)	Mean dive time (min) (SE); n = # of dives	
All krill aggregations not at sighting	NA/2019	47.31 (±0.71)	NA	
All sightings	32/892	38.09 (±0.92)	2.83 (±0.18); n > 248a	
All foraging sightings	15/398	32.64 (±1.27)	2.56 (±0.19); n > 137b	
Sightings with surface lunge feeding observed	6/122	26.42 (±2.01)	1.77 (±0.07); n = 80	
Notes.

a The number of dives monitored to assess mean dive time was not maintained for four out of 32 blue whale sightings, although the mean dive time at all sightings was recorded. Therefore, a minimum sample size of dives monitored (n = 248) is provided.

b The number of dives monitored to assess mean dive time was not maintained for two out of 15 foraging blue whale sightings, although the mean dive time at all sightings was recorded. Therefore, a minimum sample size of dives monitored (n = 137) at foraging sightings is provided.

Kinematics of the UAS filmed surface lunge feeding event

An 18.69 m female blue whale—determined through photogrammetry (Burnett et al., 2019) and genetics (Barlow et al., 2018) respectively—was filmed via UAS engaged in four surface foraging events (Table 1), including a successful SLF event. Kinematic analysis of this SLF event (Event 1, Table 1) reveals dynamic maneuvers by the whale in speed, acceleration, head inclination and roll to line up the prey for maximum engulfment (Fig. 2). The ICC and reliability ratings of the four independent estimations by each co-author of the whale’s head inclination and roll in 0.5 s interval images found that roll estimates were estimated with excellent reliability, and head inclination estimates were estimated with moderate to excellent reliability (Table S2). These results are also reflected in the low CV in Fig. 2A, all of which indicate high confidence in head inclination and roll estimates.

Initially the whale is travelling fast (2.8 m s−1) 9.0 s before the strike, then slows down when she recognizes the krill patch at a distance of 20.74 m and begins her pre-lunge roll to her left side (min roll angle −27.5°) to line-up the patch. Then she performs one full fluke beat cycle while rolling onto her right side (max roll angle 110°), and reaches maximum relative acceleration of 5.8 m s−2 1 s before strike, followed by maximum downward head inclination (−15°) 0.5 s before strike just as her mouth is beginning to open. At strike, her mouth begins to open while at a maximum relative speed of 3.33 m s−1, and reaches its widest gape angle of 33° 1.5 s after strike (Fig. 2G) when maximum upward head inclination (18.75°) and maximum deceleration (−3.7 ms−2) occurs. The duration of the lunge event, from mouth opening to mouth closure, is 4.14 s. The whale then rolls back to center and descends slowly (mean relative speed 1.22 m s−1) without taking another breath until she surfaces 2:26 min after her previous breath. The whale’s surface series prior to the SLF event included breaths 54, 38 and 18 s before strike.

The UAS footage also enables observation of fluke and pectoral fin movements relative to the foraging mechanics of a SLF event. The observed whale gets up to speed with two complete fluke cycles (4 beats): the first cycle is used during pre-lunge roll maneuvering for target patch line-up; the second fluke cycle accelerates the whale toward the patch; she stops fluking at strike, and then glides down with small fluke beats starting 9.5 s after strike. The whale flares her left pectoral fin upwards 0.5 s before strike, and then lays the fin against her enlarged buccal cavity after strike; both pectoral fins remain extended to her sides as she descends like an airplane.

Predator and prey behavior

While we are unable to calculate prey patch volume in cubic meters, the estimated area of the krill patch before and after the SLF event is 45 and 32 m2 respectively, although some krill may also be under the whale after the lunge event (Figs. 2H–2I). Reduced krill patch density is visually evident after the SLF event based on differences in color and saturation (Figs. 2H–2I). Surface krill in the target patch begin to respond to the SLF predation event when the whale is 2 m away, at 0.8 s before strike, perhaps due to the pressure wave from the whale’s rostrum (krill at depth not evaluated; Fig. 2E). At this time, the whale is moving fast (∼3 m s−1), likely to overcome the rapid escape maneuvers of krill (O’Brien, 1987). The UAS footage allows observation of individual krill ‘uncoordinated tail-flips’, group ‘flash expansion’, and propagation of escape response (Fig. 2F) through the krill swarm as described by O’Brien (1987). Although the krill move in response to the predatory whale, the distance traveled is small relative to the speed and size of the whale’s gape. Indeed, the krill do not appear to flee in an outward direction from the patch and the general shape of the patch, including the long “tail”, is maintained throughout the lunge (UAS videos available in the Supplemental Information). This lack of prey dispersion likely allows a high capture percentage for the lunging whale.

During all four UAS filmed events, the blue whale appears to use her right eye to gain visual information about krill patch size, density and orientation to inform her decisions (Fig. 4; UAS videos available in the Supplemental Information). For example, prior to the whale’s right side roll for the SLF event, she performs a pre-lunge roll to her left allowing her right eye to observe the krill patch at a distance of 18.46 m 6.5 s before strike (Fig. 2D).

Figure 4 Still images from UAS video of three separate surface blue whale foraging events captured when it was estimated that the whale visually detects the krill patch with her right eye.

Measured distances between the eye and closest edge of krill patch are given for event 1 (A), event 2 (B) and event 3 (C). See Table 2 for details.

Discussion

Based on our comparative analysis of echosounder prey data and the corresponding distribution of blue whales, it appears that New Zealand blue whales during the study period responded to the distribution of their prey by preferentially foraging in habitat with dense, surface-oriented krill aggregations. Furthermore, the short dive times of New Zealand blue whales reflects this shallow krill distribution and contrasts dive times reported for blue whales feeding in southern California (9.8 ± 1.8 min) where krill aggregations are much denser at depth (Goldbogen et al., 2011). These results indicate that blue whales in the STB of New Zealand take advantage of shallow krill aggregations, similar to the findings of Schoenherr (1991) that documented increased density of surface krill swarms near surface foraging blue whales in Monterey Bay canyon, CA, USA. Results from both studies demonstrate that foraging blue whales minimize the energetic expense of diving and oxygen consumption when possible, thus aligning with the theory that diving predators should forage close to the surface when possible (Kramer, 1988). Therefore, comprehensive assessments of blue whale foraging ecology and energetics in ecosystems with surface krill aggregations should include surface foraging behavior. Although our sample size of UAS filmed foraging events is very low ( n = 4 events), video analysis demonstrates the utility of this observation and data collection method for documentation of blue whale surface behavior. The kinematic results from the UAS video analysis of the SLF event are comparable to data derived from accelerometer and camera tags deployed on blue whales feeding at depth, and provides rare information, such as krill response, change in prey patch size after a SLF event, and potential use of visual cues to inform foraging decisions.

Compared to published speed profiles of blue whale lunges at depth derived from tag deployments in southern California (Goldbogen et al., 2011), our kinematic analysis of the UAS filmed SLF event by a New Zealand blue whale produced estimates that fit within the average range of approach speed, strike speed, and post-strike speed. Additionally, our results illustrate that peak speed coincided with mouth opening (strike), and peak deceleration coincided with maximum gape, both of which match camera tag-derived results (Cade et al., 2016). These kinematic similarities are interesting considering the smaller size of the New Zealand blue whale, relatively small gape opening used (33° compared to ∼80° for five southern California blue whales; (Potvin, Goldbogen & Shadwick, 2010; Goldbogen et al., 2011), and potentially different drag forces due to the whale’s body exposure to air after the SLF, body buoyancy, and depth (Williams et al., 2000; Goldbogen et al., 2006; Vennell, Pease & Wilson, 2006). Although our minimal sample size must be emphasized (n = 1), our estimated body kinematics of the UAS filmed SLF event generally align well with descriptions of lunge maneuvers by tagged rorquals at depth (Goldbogen et al., 2006; Goldbogen et al., 2011), including discontinuation of fluking after strike and descriptions of pectoral fin movements during the SLF event that may facilitate fine-scale steering toward the prey patch (Cade et al., 2016; Segre et al., 2018).

The UAS footage enables unique observation of krill prey response to a whale predator. We provide the first quantitative description of krill response distance and time to a foraging whale, and we can qualitatively observe the krill evasion tactics as described by O’Brien (1987). It has been theorized that larger rorquals are less maneuverable and thus require higher attack speed to capture krill before their escape response (Potvin, Goldbogen & Shadwick, 2010; Goldbogen et al., 2012). This relationship is illustrated by the greater krill response distance during Event 4 (3.77 m) when the whale was moving slower (2.69 m s−1) than the SLF event (Table 1). The barrier between the water’s surface and air may play a role in rorqual SLF as krill response into air may be different than in water. Based on the estimated response speed of dense surface schools of N. australis by O’Brien (1987) of 20 cm s−1, we can extrapolate our finding that krill flee 0.8 s prior to strike (whale’s mouth) to estimate that only krill within 16 cm of the edge of the whale’s gape would escape. We do not know the density of the targeted prey patch so we cannot estimate the amount or percent of krill captured or escaped. However, this basic extrapolation demonstrates how continued measurements of whale predation kinematics and prey response, through UAS, tags, and other technologies, can facilitate improved models and understanding of predator efficiency and predator-prey dynamics.

While it remains equivocal how rorquals detect prey aggregations at fine-scales (Torres, 2017), it has been hypothesized that vision informs foraging roll maneuvers (Goldbogen et al., 2013a) and choices at scales <30 m (Torres, 2017), and that a whale’s right eye is preferentially used to coordinate maneuvers through the left hemisphere of the brain (Friedlaender et al., 2017). The UAS footage shows evidence for all three hypotheses. Just prior to the SLF event, the blue whale performs a pre-lunge roll to her left that allows her right eye to observe the krill patch. With this visual information about prey structure, the whale then makes kinematic maneuvers to optimize her krill strike vector to maximize prey engulfment, such as orientating her lower jaw with the dense upper edge of the patch, modulating her gape size to conserve locomotor costs, and timing her jaw opening at max speed to reduce startle response in the krill. Additional sensory information may also have been used, potentially including passive listening to krill-produced noise (Torres, 2017) and tactile senses from vibrissae on the mandibles at close range (Torres, 2017). However, the whale appears to use her right eye during all four UAS filmed events, which demonstrates the importance of visual cues while foraging in these conditions and the whale’s ability to make quick behavioral choices likely based on krill patch size, orientation, distance and density (Table 1).

At all depths, rorquals perform lunges with various approach maneuvers (Goldbogen et al., 2006; Goldbogen et al., 2013a; Kot et al., 2014; Friedlaender et al., 2017), including ‘regular lunges’ where the dorsal side faces the surface, and ‘lateral lunges’ that include a body roll to one side (Goldbogen et al., 2006), such as observed in the UAS filmed whale. A whale’s choice to perform a lateral lunge at the surface may be due to good visual detection of a prey patch, allowing the whale to orient its jaws so that the largest mouth dimension captures the maximum patch. If patches are larger at depth where visual cues are reduced, mouth orientation relative to the patch may be more challenging and not as critical, making regular lunges, perhaps using a wider gape, more common. Although it has been proposed that blue whales are more likely to roll to their left when feeding at shallow depths (Friedlaender et al., 2017), we suggest that the UAS filmed whale chose to roll to her right so that the bend and orientation of the whale’s body during the lateral lunge corresponded to the prey patch. While eyeing the krill patch 6.5 s before strike, the whale could have continued on her left side, but this position would have led to an oblique intersection with the krill patch and thus not maximized prey engulfment. Hence, pre-lunge rolls may inform questions of lateralization (Friedlaender et al., 2017), and patch orientation to the whale may be a driver of roll direction, at least when targeting shallow patchy prey with good visual cues.

Blue whale surface feeding has lower energetic costs than feeding at depth due to differences in active metabolic rates (AMR), shorter transit cost between prey patches and the surface, and shorter dive recovery times (variation in drag forces by water depth are not accounted for; Goldbogen et al., 2011). Hence, when krill density is greater near the surface, such as in the STB during our 2017 survey (Fig. 3), we should expect blue whales to forage more frequently near the surface in order to optimize foraging efficiency. Even if surface krill patches are less dense, surface feeding may still be energetically preferable over diving to feed on denser patches due to relative energy gain (Doniol-Valcroze et al., 2011; Hazen, Friedlaender & Goldbogen, 2015). Goldbogen et al. (2011) estimates that a 22 m long blue whale feeding on low krill density of 0.15 kg m−3 will have an energetic efficiency of 7.9 (calculated as the ratio of energy gained from ingested krill divided by all energy expenditures of the foraging dive including mechanical energy demands, AMR, recovery time at the surface, and food assimilation costs relative to krill density). Based on the smaller length of the UAS filmed whale (18.69 m) and that lunge feeding costs scale with body size of rorquals due to mass specific engulfment capacity (Potvin, Goldbogen & Shadwick, 2010; Goldbogen et al., 2012), the foraging efficiency of the UAS filmed SLF event was likely higher than 7.9. Furthermore, the fact that the whale did not respire immediately after the SLF event indicates the low energetic cost of the lunge. However, not all surface krill patches are energetically efficient foraging opportunities, as demonstrated by Event 4 filmed by the UAS where the whale clearly decides not to engage in a surface foraging lunge based on her perception of prey patch characteristics (i.e., density, orientation), illustrating that blue whales maximize resource gain relative to both prey availability and prey accessibility.

Conclusions

This study supports the hypothesis that surface feeding is an important component of New Zealand blue whale foraging efficiency and we illustrate the utility of UAS to document these surface behaviors. To fully understand how blue whales optimize foraging effort, surface feeding events should be recorded and analyzed in conjunction with behaviors at depth, particularly in regions where prey distribution patterns are unknown. Globally, blue whale populations exhibit variation in morphology, behavior, and target prey species, which necessitates diversification of blue whale study populations and ecosystems to obtain more complete knowledge of blue whale foraging ecology. Blue whales have a large range of body length across sub-species (18–30 m), show behavioral plasticity when foraging to balance krill consumed with energetic and diving costs, and have maneuverability and foraging depth limitations that scale inversely, and allometrically with body size (Potvin, Goldbogen & Shadwick, 2010; Goldbogen et al., 2012; Goldbogen et al., 2017). Through comparative studies of blue whale foraging strategies associated with different prey ecology, ecosystems and functional body forms, we may improve our ecological understanding and management capability. For instance, blue whales in Chile are 18.9 m to 22.1 m (Durban et al., 2016), feed on a different krill species (Buchan & Quiñones, 2016), have relatively high SLF rates (Buchan & Quiñones, 2016), and demonstrate shallow diving patterns (<50 m; Bocconcelli et al., 2016). Moreover, such diversity in foraging strategies may help explain the evolutionary diversification of blue whale sub-species by better linking form and functional ecology, such as the relatively small New Zealand blue whale that potentially sacrificed engulfment size and diving capacity for increased agility to capture patchier surface prey.

Using UAS as our observational platform, we estimated the kinematics and prey response of a blue whale’s surface foraging events—behaviors that can be difficult to quantify in accelerometer or camera tag data. We analyzed our UAS video footage of just four surface foraging observations by one whale as a proof of concept, and were able to provide new information on New Zealand blue whale foraging kinematics, energetics, sensory cues, and predator-prey response dynamics. We suggest that incorporating UAS into studies of blue whale foraging will complement tagging data by providing information on surface feeding rates, kinematic coordination of fluke, pectoral fin, and mouth movements, and, when conditions are favorable, the prey patch. UAS is a minimally invasive data collection technique (Christiansen et al., 2016; Domínguez-Sánchez, Acevedo-Whitehouse & Gendron, 2018), and with increased replicate observations, it could be an excellent tool to enhance our understanding of rorqual surface foraging ecology and prey patch geometry and response to predation. Indeed, a recent study applied GoPro cameras and UAS to improve descriptions of humpback whale oral morphology during surface feeding (Werth et al., 2019). Although the current UAS battery life is typically limited to ∼20 min, operation of multiple UAS to alternate flight time with battery swaps could facilitate continuous recording. Simultaneous measurements of prey and predator at fine-scales during feeding events are rare but critical to understanding ecological processes and foraging efficiency (Goldbogen et al., 2015), and whale-borne video tags (Calambokidis et al., 2007; Cade et al., 2016) and UAS can fill this data gap. While more studies are needed to understand the complex interplay between blue whales and their prey (Goldbogen et al., 2013a), we contend that application of different technologies in a broader range of ecosystems would be highly beneficial.

Supplemental Information

Table S1 Mean and standard deviation of measured features for all krill aggregations (n = 2, 911)

Click here for additional data file.

Table S2 Results of the Interrater Correlation Coefficient (ICC; (Koo & Li, 2016)) of consistency and agreement between the four independent co-author evaluations of the blue whale’s head inclination and roll estimates of 0.5 sec interval images

F-test conducted with a 2-way mixed-effects model.

1Scores of groups of observations are correlated in an additive manner among raters.

2Different raters assign the same score to the same observation.

Koo TK, Li MY (2016) A Guideline of Selecting and Reporting Intraclass Correlation Coefficients for Reliability Research. Journal of Chiropractic Medicine 15:155-163

Click here for additional data file.

Figure S1 Pairwise comparisons of density, depth, and thickness of each krill aggregation (n = 2,911)

Plots on the diagonal show the distribution (probability density) of each metric. Plots in the upper right list the Pearson’s correlation coefficient between each pair, and scatter plots in the lower left show the relationship between each pair.

Click here for additional data file.

Figure S2 Example images extracted from UAS video of blue whale surface lunge feeding event to calculate blue whale body roll

Each 0.5 sec image was evaluated by the 4 co-authors and the mean value is given with each image and time stamp.

Click here for additional data file.

Supplemental Information 5 Full UAS videos and krill aggregation data files analyzed

Click here for additional data file.

The project was accomplished through the dedicated fieldwork of many individuals including the crew of the RV Star Keys (Western Work Boats, Ltd.), Kristin Brooke Hodge, Mike Ogle, and Callum Lilley. We are grateful to Kim Bernard and Pablo Escobar-Flores for assistance with echosounder data analysis, to Kim Bernard and Daniel Palacios for constructive feedback on this manuscript, and to David Cade and two anonymous reviewers for highly constructive and insightful reviews that improved this manuscript.

Additional Information and Declarations

Competing Interests

Author Contributions

Animal Ethics

Field Study Permissions

Data Availability

The authors declare there are no competing interests.

Leigh G. Torres conceived and designed the experiments, performed the experiments, analyzed the data, prepared figures and/or tables, authored or reviewed drafts of the paper, and approved the final draft.

Dawn R. Barlow performed the experiments, analyzed the data, prepared figures and/or tables, authored or reviewed drafts of the paper, and approved the final draft.

Todd E. Chandler performed the experiments, authored or reviewed drafts of the paper, and approved the final draft.

Jonathan D. Burnett analyzed the data, authored or reviewed drafts of the paper, and approved the final draft.

The following information was supplied relating to ethical approvals (i.e., approving body and any reference numbers):

Data collection on blue whale ecology used in this study was non-invasive and hence we received an exemption from Oregon State University IACUC (Exempt 16-1083).

The following information was supplied relating to field study approvals (i.e., approving body and any reference numbers):

Field work on blue whales was approved by the New Zealand Department of Conservation (# 45780-MAR).

The following information was supplied regarding data availability:

Data are available in Figshare: https://figshare.com/projects/Insight_into_the_significance_and_kinematics_of_blue_whale_surface_foraging_through_drone_observations_and_prey_data_Supplementary_Information_/74127.

The full Unmanned Aerial Systems (UAS) videos of the blue whale surface lunge feeding event and other surface events analyzed in this study can be viewed in the Figshare repository. Please refer to the time stamps in Table 1 of the manuscript to view the four foraging events described. The footage was filmed by Todd Chandler, and ownership of UAS videos belongs to Leigh Torres, Oregon State University. These UAS videos should only be used for scientific purposes and should not be shared on social media or broadly without explicit permission from L. Torres: Torres, Leigh; Barlow, Dawn (2020): UAS blue whale videos. figshare. Media. https://doi.org/10.6084/m9.figshare.11595246.v1.

Krill aggregation data files examined in this study (.csv files) and the Matlab script used to identify aggregations and their attributes are available at Figshare. Additionally, data on krill aggregation characteristics at blue whale sightings and at absence locations, and the R script used to analyze and plot these data, are available:

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
