# Peer review of "Insight into the kinematics of blue whale surface foraging through drone observations and prey data"

_PeerJ, doi:10.7717/peerj.8906_

## Round 0.1 · original submission · Major Revisions

The three reviewers and I agree that this is an incredible observation that provides valuable data on krill distribution and whale foraging behavior. There are three main points brought up by the reviewers that need attention in the revision.

1. We need more information on how the kinematic measurements were made and specifically how you controlled for the movement of the UAS platform relative to the movement of the whale.
2. The reviewers highlight the value of quantifying the prey escape response.
3. The reviewers feel that you overstated the significance of this observation relative to the prior literature on blue whale foraging, and especially the importance of surface vs. deep foraging. Please revise the ms to remove unnecessary statements of novelty. Note, from the PeerJ acceptance criteria: "Decisions are not made based on any subjective determination of impact, degree of advance, novelty..."

I look forward to reading your revision.

·

Basic reporting

Generally, this manuscript does a good job of describing results and placing them in the literature. It includes a rarely captured overhead view of a blue whale surface feeding event and interestingly documents some measure of prey response. I think the authors actually undersell the importance of the prey escape response they recorded and could even provide additional details on these findings (at the authors’ discretion of course but could strengthen the manuscript, see final comment). This could be an important contribution to the literature. The manuscript also provides an interesting characterization of the prey field in this environment (though see comments below) that suggests that in this environment blue whales may depend more on shallow prey aggregations, increasing their overall efficiency.

I do think, however, that some of the language in other places regarding the results being “unique” (e.g. line 361, 416 etc.) and “novel” (e.g. line 400) is perhaps a little overzealous given the number of UAV studies of cetaceans (e.g. Johnston 2019, full cite below) and the number of on-animal video observations reported of blue whale lunge feeding that have provided similar insights. Not suggesting you need to cite these (especially given that I am pointing towards several of my own papers here), just providing for your reference and suggesting to remove “unique”:
Segre, P. S., D. E. Cade, J. Calambokidis, F. E. Fish, A. S. Friedlaender, J. Potvin and J. A. Goldbogen. 2018. Body flexibility enhances maneuverability in the world’s largest predator. Integrative and comparative biology.
Cade, D. E., A. S. Friedlaender, J. Calambokidis and J. A. Goldbogen. 2016. Kinematic Diversity in Rorqual Whale Feeding Mechanisms. Current Biology 26:2617-2624.
Segre, P. S., S. M. Seakamela, M. A. Meÿer, K. P. Findlay and J. A. Goldbogen. 2017. A hydrodynamically active flipper-stroke in humpback whales. Current Biology 27:R636-R637.
Gough, W. T., P. S. Segre, K. Bierlich, D. E. Cade, J. Potvin, F. E. Fish, J. Dale, J. Di Clemente, A. S. Friedlaender and D. W. Johnston. 2019. Scaling of swimming performance in baleen whales. Journal of Experimental Biology:jeb. 204172.
Werth, A. J., M. M. Kosma, E. M. Chenoweth and J. M. Straley. 2019. New views of humpback whale flow dynamics and oral morphology during prey engulfment. Marine Mammal Science.
Johnston, D. W. 2019. Unoccupied aircraft systems in marine science and conservation. Annual review of marine science.

Additionally, while the authors have clearly made effort to compare their result to previous literature, I think, given an n of 1, some of this discussion is not really warranted (e.g. line 408-412). Speeds in the other papers that you are comparing to are all averages of many lunge feeding events. Your observation is not really abnormal given the normal variation in blue whale lunge feeding (see, for a clear example, the means and st. dev. listed in Table 1 of Cade et al. 2016 cited above), so without further information it’s hard (and not really necessary) to speculate as to whether the lunge you measured is actually that different from other reported values and what the reasons could be (since it’s not statistically different). Better are times where you remark that the observations of speed and engulfment in your video fits into reported values.

Finally, I would recommend scaling down the mentions regarding the lack of surface feeding in the literature. It’s not that studies haven’t considered it (you cite Goldbogen 2011, for instance), but surface feeding is often excluded from other studies because of a) limited hydroacoustic prey data for appropriate comparisons and b) biomechanical processes and forces change when half the whale is out of the water. This should be added to discussion in line 102 etc. Some of examples of papers that directly tackle surface feeding include:

Allen, A. N., J. A. Goldbogen, A. S. Friedlaender and J. Calambokidis. 2016. Development of an automated method of detecting stereotyped feeding events in multisensor data from tagged rorqual whales. Ecology and Evolution 6:7522-7535.
Doniol-Valcroze, T., V. Lesage, J. Giard and R. Michaud. 2011. Optimal foraging theory predicts diving and feeding strategies of the largest marine predator. Behavioral Ecology 22:880-888.
Iwata, T., T. Akamatsu, S. Thongsukdee, P. Cherdsukjai, K. Adulyanukosol and K. Sato. 2017. Tread-water feeding of Bryde’s whales. Current Biology 27:R1154-R1155.
Kot, B. W., R. Sears, D. Zbinden, E. Borda and M. S. Gordon. 2014. Rorqual whale (Balaenopteridae) surface lunge‐feeding behaviors: Standardized classification, repertoire diversity, and evolutionary analyses. Marine Mammal Science 30:1335-1357.
Owen, K., R. A. Dunlop, J. P. Monty, D. Chung, M. J. Noad, D. Donnelly, A. W. Goldizen and T. Mackenzie. 2016. Detecting surface‐feeding behavior by rorqual whales in accelerometer data. Marine Mammal Science 32:327-348.

Experimental design

No comments, though see notes on acoustic analysis below.

Validity of the findings

Most of the manuscript seems to have a solid foundation. I wish the error in speed and distance measured from the UAV were better quantified given the motion of the UAV, but I do not believe those overly distract from the results.

The krill density analysis, however, is not sufficiently described such that a reader can follow the procedures, and there seems to be at least one potentially major mischaracterization of the data.

*keeping this paragraph here, but see note in next paragraph* On line 210 the authors describe their method for isolating krill, but the numbers and procedures are not well described. Line 210 is very confusing and seems to suggest the authors excluded all data above -90dB? In almost any ecosystem and acoustic set up this would essentially be all of the data (assuming it is a calibrated system, though the calibration procedure was not described but should be). -90 dB is like a single krill / m3 (or 10 krill if the krill are very small), this is basically background noise and suggests your analysis is inappropriately excluding most patches that have any kind of density. With only a single frequency transducer, there’s no real way to exclude fish species (and the cited source from Bernard and Steinberg is from a different ecosystem on a much larger species of krill, and their method is not especially robust and should be used with caution given their single frequency). The described method would need a lot more justification and additional details, including an understanding of the target strengths of other types of organisms that may be returning echos, currently it sounds like lots of data may have been inappropriately excluded. Especially in known foraging areas, it may be more prudent to actually leave in all data and assume it is krill (except for noise of course), given that you can’t know for sure. Or do one pass trying to differentiate (though I’m not convinced that’s possible) and one pass where nothing is excluded. Presenting both analyses would at least let the reader draw better inferences.

*I keep re-reading this description- I’m actually thinking now that you probably excluded data below -90 dB (the opposite of what you said on line 211). At least this would make sense and would exclude background noise and help identify the boundaries of schools. This method would definitely not differentiate krill from other ensonified species (like the fish example given, which have high TS), however, so some discussion should be devoted to if there was any ability to exclude fish schools (e.g. based on school shape), or at the very least what the general prevalence of such non-krill species is in this ecosystem. If this opposite reading is what was actually done, some additionally validation of the -90 dB threshold should be given (i.e. based on the target strength of the krill in your ecosystem)

A second issue in the acoustic analysis regards the depth of the patches (around line 216), it is not clear exactly what “mean depth” represents. That is, is it the mean depth of the densest part of the patch? Or just the depth of the aggregation? To give a sense of what is meant by “mean depth”, a sense of the size of the krill school is also necessary. For instance, if an aggregation is 100 m thick, centered at 50 m depth, that’s very different, in terms of availability to blue whales, than if an aggregation is 1 m thick centered around 50 m. Reporting mean thickness may help, especially since you are using NASC, which is an aggregation of the whole water column. As you point out, density is critical, so it matters a lot when reporting NASC of a patch what the thickness of the patch is. May be better to report and analyze mean volume backscattering (Sv) of the patch instead.

Additional comments

See minor comments on the following lines

Line 32- Awkward sentence, seems to imply that blue whales were encountered when surface lunge feeding was observed, which is a weird statement

Line 41-43 prefer to lay out what the observations were, rather than just saying unique observations were made

Line 68- because gray whales appear to be more firmly placed within Balaenopteridae, “rorqual family” isn’t quite correct anymore. Better to say “blue whales are rorqual whales”

Line 70, perhaps these three citations could be condensed just to the Goldbogen et al 2017 paper you cite later on.

Line 75- re: “so that calories gained outweigh foraging costs”. This is true, of course, but even low density patches < ~100 g/m^3 likely outweigh costs. Perhaps better just to say to “maximize energy gain” or something along those lines.

Line 78- remove “notoriously”

Line 94-95- the parenthetical here is not really necessary as the sentiment that krill species can affect prey quality is also encapsulated in the previous statement about more general density

Line 96-98, thoughout be a bit careful with vocabulary. It’s not actually instantaneous efficiency (defined as energy in compared to energy out) that matters for a foraging animal over time, but net energy gained over time. The calculations in Goldbogen 2011 can be used to support this (i.e. the net energy gain over 12.5 minutes of a dive is still less than assuming continuous surface lunging every ~1.5 minutes at the surface if krill density was the same). Recommend rephrasing the comparison calculations throughout

The implication that blue whales mostly surface feed in New Zealand (line 101) should be walked back a bit unless more general information can be presented. Better to say that the observed blue whales were mostly surface feeding.

Line 104- true that automatic detection methods can be confounded, but that does not mean lunges are undetectable or excluded for that reason. Human analysts can fairly accurately differentiate lunge feeding signals from surface noise.

Line 188 consider reporting the size and sex of the observed whale here.

Line 187-198, seems like a lot of unnecessary detail. Can just say one flight over a surface feeding whale was completed

Line 222- more explanation- could not be determined why?

Line 223- Are these estimates from the cited paper? If this is not directly measured data, some caveat should be stated (e.g.: “surface swarms can range between xx and yy”). Also more helpful than numbers of krill (or in addition to) should be biomass as that is the unit described elsewhere in the paper.

Line 226- 100 g/m^3. Your krill are much smaller than 1 g/ individual and likely 100 krill are not sufficient.

Line 228- this assumption seems unwarranted, but also not necessary to your other observations. Better just to say the density of the swarm was unknown.

Is there any estimate of the importance of surface feeding?

Line 242- What about pitch (as in Fig 3?)

Line 242-257 Given the uncertainties, recommend calling the quantities “relative speed” and “relative acceleration” throughout the manuscript

Line 260- was “max gape” later reported?

Line 266-270, how does the “interval of interest” here relate to the information in lines 248-252? Is it the same? Can one of these sections be streamlined?

Line 297-300- not clear how roll was calculated? Was it an estimate or was there some derivation of the values? (okay I see in line 310 that these are estimates, please list earlier)

Line 304-309- are you using “head inclination” and “pitch” interchangeably? Since pitch conventionally refers to body pitch, recommend changing to “head inclination” throughout the manuscript which is an interesting metric and worth differentiating

Line 325- Some of the language is a little strong. My understanding is that foraging behavior was presumed from surface observations of dive behavior. This can an unreliable indicator foraging, so best to say a less strong word than “confirmed” when using surface observations and presumptions of ARS.



Line 374- a “likely” is warranted here, as there are several hypotheses for why whales lunge quickly, including biomechanical based hypotheses:
Orton, L. S. and P. F. Brodie. 1987. Engulfing mechanics of fin whales. Canadian Journal of Zoology 65:2898-2907.
Potvin, J., J. Goldbogen and R. Shadwick. 2009. Passive versus active engulfment: verdict from trajectory simulations of lunge-feeding fin whales Balaenoptera physalus. Journal of the Royal Society Interface 6:1005-1025.

Line 391- It seems fairly clear from your general depiction of the preyscape that blue whales here are foraging in shallow water because the krill is in shallow water. Unless there is evidence that there is deeper krill that the whales are choosing not to forage on, it seems unlikely that they are making an active energetic choice, but instead are just foraging where the krill is.

Line 394-395- not sure this statement is necessary, or even justified here. You show that blue whales sometimes feed at the surface, but there is not much discussion about the proportion of their time spent surface feeding. A better statement might be more locally based, e.g. “blue whales in our study area appear to get a substantial portion of their diet from near-surface foraging at least during daylight hours, so new methods of analysis are available.” Or something to that effect

Line 397-399, recommend making some of the comparison explicit. Agree that your measurements are comparable to others, but worth making that explicit comparison for readers that are not steeped in the literature. An additional comparison that may further justify your findings is that the timings of engulfment, with mouth opening at peak speed and maximum gape coinciding with peak deceleration, also match findings from camera tags (Cade et al. 2016). Do not have to cite this paper, but may further strengthen your results.

Line 407- Unclear how speed “allows” lower mass-specific engulfment capacity? I think you are looking for a different word there.

Line 415- “except that the whale did not fluke through the lunge”. My understanding is that a basic assumption of most of the models in the papers cited assume that fluking stops at mouth opening, so in that sense your results match.

Line 418- agree. These results could be further highlighted. The reaction distance of the krill is a great metric to report. To make the results even more meaningful, the authors could consider including (not necessary, just a suggestion) a brief calculation about the % of the krill patch that could escape given the observations of when they start to react and the observed escape speeds around the edges. Even if all krill escape outwards away from the whale, given the speed differential between the krill and the whale, the response time (~0.8 s from your findings) and the scale of the whale’s mouth, this escape proportion is assumed to be small, but would be very useful for future models of efficiency to state explicitly the number that you would find.

Reviewer 2 ·

Basic reporting

No comment

Experimental design

No comment

Validity of the findings

I still need to be convinced that kinematic measurements can be conducted from a moving drone, otherwise the specific kinematic results should be omitted.

Additional comments

This study describes and analyzes the surface lunge feeding behavior in a population of Pygmy Blue Whales (Balaenoptera musculus brevicauda) and contends that this behavior is key to understanding the ecology and energetics of this species. I commend the authors for their efforts, which will aid in our understanding of this iconic species and SLF behavior. Furthermore, having high-resolution video evidence of predator-prey patch behavior is uncommon and should be published as it lends to testable predictions on predator prey dynamics, schooling/swarming behavior, and sensory ecology of both predator and prey. However, I have several major issues that need to either be addressed or re-worded before I can endorse this study for publication. All the changes I recommend are doable however, and along with them I suggest a title change to: “Insight into blue whale surface foraging through drone observations and prey data”.

Major points:
I do not understand how it is possible to obtain specific kinematics on a moving animal subject – that is otherwise uninstrumented – from a moving drone. Typically, when kinematics are determined from video imagery the camera is still, or each still frame is calibrated to correct for the movement of the image capturing device, a drone in this case. While the kinematic details are interesting and in line with what we know from tagging studies of this species in the Eastern North Pacific, I do not see how they can be considered robust with the methods as described. For example, animal speed is notoriously tricky to measure accurately even when a tag is affixed to the animal; to get measurements as precise as a tenth of a m/s seems impossible from a drone. The same can be said for the roll, and other kinematic measures, all reported with one-degree precision. I am willing to be convinced by expanding the methods to fully explain how this is possible, but as the paper currently reads results this specific should not be reported. This does not have to be a fatal flaw of this study however; I think describing the SLF events using relative kinematics would be perfectly fine. For example, “Just prior to prey capture the animal rolled on its right side, accelerated, and lunged laterally into the patch,” and similar, would be acceptable.

This flows into my other major concern, which is the extendibility of these findings to the ecology and energetics of the blue whale as a species. The kinematics, and associated conclusions, were drawn from one observed feeding event. This is a very exciting natural history observation worthy of reporting and describing, but to claim that this is how all, or even most, SLF is in this population or this species as a whole is over-reaching what the authors’ data is able to support.

The energetic argument is flawed in that the energetic cost of transport to and from “deep” krill patches (typically 100-300m deep) is predicted to be extremely low. Further, even a patch that is 250m deep is only ~10 body lengths for a blue whale. Therefore, act of diving itself, in both time and energetic expenditure, is extremely low for a blue whale. A more parsimonious energetic explanation is that blue whales simply feed on krill wherever it happens to be densest in that time and place. Also, a better test on energetic efficiency would be to compare the feeding rates at the surface and at depth, which I understand is beyond the scope of this study, but to say that SLF is a critical component of blue whale ecology it is critical to look at the whole picture. I am confident that the authors’ have conducted enough observations to state that SLF is an important aspect of the feeding ecology of this population, but to say that it is important to the overall foraging ecology of this cosmopolitan species is over-reaching. There are very few behaviorally well-studied populations of blue whales, perhaps the only other one is the Eastern North Pacific population, where, as you reference, SLF appears to be a relatively uncommon (but not unheard of or unseen) behavior. It is fair to say that for this population of pygmy blue whales SLF is an important (and perhaps previously unconsidered?) component of their foraging ecology. At present, there is not enough rich behavioral data available for the other blue whale populations to speak to the importance of SLF overall.



Minor points:
• Line 226: Your talking about prey density but then you provide a volume of water. Perhaps you meant the spatial extent of a prey patch and not it’s density?
• Line 255: Add space before citation.
• Line 415-416: Rorquals typically do not fluke through the lunge, doing so would crater their energetic efficiency.
• Line 461: Add space after semicolon
• Lines 470-474: There is not enough data to conclude the specific value of this particular whale’s energetic efficiency; see major comments above.

Reviewer 3 ·

Basic reporting

It is always fantastic to have new data for animals in their natural environment. The authors provide new prey data in blue whale habitat in New Zealand. Although blue whales are relatively well studied in other areas, little is known about blue whales in NZ. Specifically, new information on the distribution of krill in relation to blue whale sightings at the sea surface is the primary strength of the manuscript. However, there are many weaknesses of the manuscript. In particular, one aspect of the study (kinematics) is technically flawed and should be removed.

As the title indicates, the manuscript aims to provide insight into two aspects of blue whale foraging: 1) significance and 2) kinematics. I will focus my review on each.

Experimental design

See other fields for comment.

Validity of the findings

1. Significance. The authors are making the claim that surface lunge feeding is important and that prior studies disregarded this part of their foraging ecology. This latter is is patently false for many reasons (see a, below). For the former, surface lunge feeding may very well be important, but the data in the paper cannot speak to whether it is or not. For example, the authors were not able to measure any potential feeding at depth, so any feeding rates of surface versus depth cannot be assessed with this data.

a) Many studies have looked at surface lunge feeding in rorquals, and also blue whales more specifically (Doniol-Valcroze
et al., Kot et al.). In particular, Jill Schoenherr’s study (1991, Can J Zool) on deep and surface krill patches in blue whale forging habitat is the most comprehensive study to date that compares deep and shallow. This is a major omission and needs to be addressed. Therefore, it is not clear how the research fills an identified knowledge gap.

b) The authors fail to define what significance is, as well as how it would be measured and statistically compared. A dichotomy between surface and deep feeding is present in the narrative of the paper, but it is unclear what would constitute significance in this context. A certain proportion of lunges at the surface versus shallow? This needs to be defined. If a lunge occurs at 5 or 10 meters in depth is it still a surface lunge? Is there a arbitrary or justified threshold that distinguishes the two categories or is it a continuum? At what depth will UAS not be able to view the whale? In summary, the research question is not well defined in the manuscript, and it is not clear how the data are relevant & meaningful in the context of blue whale foraging ecology.

2. Kinematics. The kinematic analysis is technically flawed and should be removed. The authors should report the amazing behavior observed by the UAS, but attempts to quantify kinematics are problematic because the UAS was moving, currents were not accounted for, and krill may undertake primary escape responses in well lit environments (O’Brien, 1987). All these factors complicate any kinematic analyses with respect to speed and acceleration. Also, it is not possible to measure roll as a function of time from one overhead camera. When I looked for methods on roll estimation, it is not described in sufficient detail for replication in the manuscript.

---

## Round 0.2 · Minor Revisions

Please respond to Reviewer 1's feedback on the use of NASC as a metric of prey density.

·

Basic reporting

Overall, I thought the authors did a good job addressing reviewers’ concerns and I found the manuscript much more of a compelling read. I appreciated the authors attention to language and to the reviewers’ comments. There is a major issue with the prey analysis (discussed in detail in the “Validity of the findings” section below) that needs to be addressed before the manuscript can be accepted, and generally the description of the prey analysis could be much tighter and clearer, but other comments are small.

General comments:
I appreciated the increased focus on error with regards to kinematics. Unless I missed it, I didn’t not see a similar error reported for whale speed/distance. Some discussion of the potential error in speed estimation (in numerical form, even if an additional estimation) given the different moving pieces seems warranted (in addition to the helpful caveats given describing why the error exists).

Thank you for including the raw video that allows the reader to examine the escape responses laid out in lines 397-407. After viewing the video, it is perhaps worth an extra sentence at the end of this paragraph noting that the krill appears to move around in response to the whale, but do not appear to move in an outward direction away from the whale, so capture percentage is likely high. Indeed, the general shape of the patch (including the off-shoot arm on the right side of the frame) appears to be maintained throughout the lunge.

Line 499-500. It is worth a little bit more explanation of why surface feeding energetics are presumed to be lower than deep feeding. Wave drag is higher at the surface, so it’s my understanding that the actual cost of lunging is not smaller at the surface, but the overall reduction in cost is more likely related to the shorter recovery times (from the shorter dives you describe) as well as shorter transit times.

Prey comments-

Generally, it was a little difficult to know exactly what steps were taken in the analysis, but I do thank the authors for including well-commented, easy-to-follow code that allowed me to make the following observations that should improve the descriptions in the methods, but also allowed the identification of a major issue that, when fixed, should give more convincing results.

Minor prey comments:
NASC as a metric of density. As I mentioned in the first review, NASC is typically used in two dimensions as it integrates the vertical dimension to measure abundance over an area (as opposed to density in a patch). These authors use NASC to measure changes in the vertical domain, in which case care should be taken to justify why this is appropriate. The explanation should highlight that because 1 m vertical bins were used, NASC and sv essentially only differ by a multiplicative factor (i.e. NASC/(4pi*1852^2), see Maclennan reference below). I would still suggest using sv (or better, Sv, see below) as that additionally standardizes for bins of unequal size (i.e. some bins in the dataset are ~ 1.1 m and some are 0.9, meaning that a 1.1 m bin would have 20% more backscatter than a 0.9 m bin even if density was the same when using NASC; this problem is ameliorated by using sv), but those differences should not substantially affect results. If NASC is used, places to further explain how it was used would be in lines 227-228, for example, where the authors could emphasize that the NASC of the 1m thick bins were averaged within krill swarms (i.e. it is not clear right now from that description if the NASC of the whole swarm itself was calculated, which would give the areal vs density problem, or if the 1 m x 5 ping bins were averaged).

Sv vs sv. Throughout the manuscript, these appear to be used interchangeably (see, e.g., Fig S1). In conventional usage, Sv is the logged form of sv, that is Sv = 10 log10 (sv). See the guide to definitions and symbols here:
MacLennan, D.N., Fernandes, P.G. & Dalen, J. (2002) A consistent approach to definitions and symbols in fisheries acoustics. ICES Journal of Marine Science, 59, 365-369.
This will be critical later, as improper averaging of Sv in the logarithmic domain is one of the problems identified in the code.

The term “patch width” is used throughout the manuscript, but I think from the usage in the code the authors are referring to patch size in the z-direction, so “patch thickness” may be a clearer term to use as width is often a dimension along the x and y axes.

Supplemental figure S1 is hard to interpret. Would it be possible to place the correlations on the plot that they go with? Looking in two disparate places makes it challenging to draw connections. A legend/caption would help as well, though I do not see one in the files available to reviewers. I’m not familiar enough with these types of plots to know what, for instance, a plot of mean depth vs mean depth with different scales means.

Line 228-229. The description of how depth was calculated was a little hard to follow. Perhaps say more explicitly that mean depth was calculated as the mean depth of the patch midpoints. Another suggestion would be to calculate the depth of the densest bin in each vertical column, but regardless a clearer explanation would be useful.

Experimental design

It should say explicitly in the text what surface values were excluded. Your spreadsheets have very high values in the first m, which is in the near field of the transducer so need to be excluded or they will bias your results. Code suggests first two m were excluded (line 142), so just state that this was the method applied to all data

What software was used for the original export of acoustic data to csv file?

When/how was the echosounder calibrated? If uncalibrated, important to state.

Validity of the findings

There is a major issue in the prey analysis that will lead to large underestimation of patch density in some cases that needs to be corrected.

This issue in analysis appears in line 296 “nasc = mean(mean((NASC_Agg(minr:maxr,minc:maxc))))”
as well as line 300 of the matlab code. Earlier in the code, empty water was replaced in the aggregation matrices “Krill” and “NASC_Agg” with zeros. But then in line 296 those zeros are averaged into overall density estimation (since the code as written takes the mean of the columns, and the means of those values) which will bias your values low (if using NASC, high if using Sv), particularly for patches with both thin and thick sections. See figure below (included in the pdf attachment to this review) which is the 3rd aggregation in the first csv file. In this example, averaging as is done in the code results in a value that is half what the actual value should be.


Instead, you need to find the mean of just the patch. So code for you would look instead like this:
NASC_Agg(find(Aggs==i))
Which gives you a column vector of just the bins within the patch (without the zeros). Taking the mean of this will give you the mean NASC of the aggregation. There is a second reason why this is better as well. Even if you removed the zeros, the way the code is now, “mean(mean(…” takes the mean of the columns and then averages those. This will create a bias towards the thin parts of the patch. Taking the mean of all the data combined (i.e. “mean(NASC_Agg(find(Aggs==i)))” gives you the overall mean of the patch.

Additionally, line 300 of the code involves the “Krill” table, which are Sv values. The same problems from above apply, but additionally, if you are trying to calculate a mean patch density, you need to average values in the linear domain (i.e. sv). NASC is already in linear domain (which is why you log it for comparisons and figures), but Sv, which is in decibels, needs to be converted back to sv before averaging (see equation above and Maclennan et al. cited above), and then converted back to Sv after averaging.

Additional comments

no comment

Reviewer 2 ·

Basic reporting

no comment

Experimental design

no comment

Validity of the findings

no comment

Additional comments

I am happy with the changes made by the authors. The revised manuscript has the results limitations clearly stated and better contextualizing the nature of their observations. In the long run this will help science build off of and improve on this work. I have only one minor comment for the authors this round.

My main comment for this version is I am not sure about the use of the word "relative" to describe the kinematic measurements. It leaves me thinking "relative to what?" Perhaps a better way phrase this would be "estimated", "approximated" or similar. I don't feel very strongly about this, rather that the use of the world relative here seems odd.

---

## Round 0.3 · accepted · Accept

Thank you for your through response to the reviewer's feedback. I believe the review process has been quite helpful and I'm very happy with the final manuscript.